# Evaluation of a Single Determination of Gluten Immunogenic Peptides in Urine from Unaware Celiac Patients to Monitor Gluten-Free Diet Adherence

**DOI:** 10.3390/nu15051259

**Published:** 2023-03-02

**Authors:** Vincenza Lombardo, Alice Scricciolo, Andrea Costantino, Luca Elli, Giorgia Legnani, Ángel Cebolla, Luisa Doneda, Federica Mascaretti, Maurizio Vecchi, Leda Roncoroni

**Affiliations:** 1Fondazione IRCCS Ca’ Granda Ospedale Maggiore Policlinico, 20122 Milan, Italy; Center for Prevention and Diagnosis of Celiac Disease; Gastroenterology and Endoscopy Unit; vincenza.lombardo@policlinico.mi.it (V.L.); alice.scricciolo@policlinico.mi.it (A.S.); andrea.costantino@policlinico.mi.it (A.C.); giorgia.legnani@studenti.unimi.it (G.L.); federica.mascaretti@policlinico.mi.it (F.M.); maurizio.vecchi@unimi.it (M.V.); leda.roncoroni@unimi.it (L.R.); 2Department of Pathophysiology and Transplantation, Università degli Studi di Milano, 20122 Milan, Italy; 3Biomedal S.L., Polígono Industrial Parque Plata, Calle Calzada Romana 40, Camas, 41900 Sevilla, Spain; acebolla@biomedal.com; 4Department of Biomedical, Surgical and Dental Sciences, Università degli Studi di Milano, 20122 Milan, Italy; luisa.doneda@unimi.it

**Keywords:** celiac disease, gluten immunogenic peptides, gluten-free diet, gluten-free diet monitoring, urinary gluten detect

## Abstract

Introduction and aim: Usually, adherence to the gluten-free diet (GFD) in celiac patients is indirectly assessed through serological analysis, questionnaires, or invasive methods such as intestinal biopsy. The detection of gluten immunogenic peptides in urine (urinary gluten immunogenic peptides—uGIP) is a novel technique that directly evaluates the ingestion of gluten. The aim of this study was to evaluate the clinical efficacy of uGIP in the follow-up of celiac disease (CD). Methods: From April 2019 to February 2020, CD patients reporting complete adherence to the GFD were prospectively enrolled but were unaware of the reason for the tests. Urinary GIP, the celiac dietary adherence test (CDAT), symptomatic visual analog scales (VAS), and tissue transglutaminase antibodies (tTGA) titres were evaluated. Duodenal histology and capsule endoscopy (CE) were performed when indicated. Results: A total of 280 patients were enrolled. Thirty-two (11.4%) had a positive uGIP test (uGIP+). uGIP+ patients did not show significant differences in demographic parameters, CDAT, or VAS scores. The tTGA+ titre was not related to the positivity of uGIP (14.4% vs. 10.9% in patients with tTGA+ and tTGA−). Regarding histology, 66.7% of the GIP+ patients had atrophy compared to 32.7% of the GIP patients (*p*-value 0.01). However, the presence of atrophy did not correlate with tTGA. Mucosal atrophy was detected in 29 (47.5%) out of 61 patients by CE. With this method, no noticeable dependence on uGIP results (24 GIP− vs. 5 GIP+) was observed. Conclusions: The single uGIP test was positive in 11% of CD cases referring a correct GFD adherence. Furthermore, uGIP results significantly correlated with the duodenal biopsy, formerly considered the gold standard for assessing CD activity.

## 1. Introduction

Celiac disease (CD) is a systemic immune-mediated disorder triggered by gluten ingestion in genetically susceptible individuals carrying the HLA-DQ2 and/or DQ8 haplotypes. It is characterized by intestinal (small bowel) atrophy, specific autoantibodies (anti-tissue transglutaminase immunoglobulin A (tTGA), and anti-endomysial antibodies (EMA)) [1,2,3,4].

CD is currently one of the most common chronic diseases, affecting 0.5% to 1% of the general population nationwide and worldwide, except for areas with a low frequency of genetic predisposition and low gluten consumption [5,6,7].

To date, the only effective therapy for CD is a strict, lifelong GFD and it is generally accepted that strict adherence to a GFD results in serological, histological, and clinical remission. A GFD is also effective on extra-intestinal CD manifestations. These could be short stature in children, infertility, anemia, improvement in body weight, metabolic and nutritional changes such as a significant increase in bone density, and the normalization of vitamin (e.g., vitamin B12) and mineral levels. Notably, a GFD could prevent CD-associated complications: poor dietary adherence favors the development of other autoimmune diseases, fertility problems, and increased risk of bone fractures. Also, although rare, other life-threatening complications can occur including hyposplenism, refractory coeliac disease, intestinal lymphoma, small bowel adenocarcinoma, and ulcerative jejunoileitis [1].

In many cases, the persistence of symptoms despite a GFD may be attributed to other coexisting causes, such as irritable bowel syndrome, microscopic colitis, intolerance to other food ingredients (e.g., lactose, histamine), or inflammatory bowel disease. Non-responsive CD occurs when patients have persistent symptoms or when mucosal damage is present despite an apparently adequate GFD [8]. 

The persistence of symptoms or intestinal mucosal damage may also be triggered by continued and involuntary gluten ingestion. The frequent daily gluten exposure makes the GFD very difficult to accomplish. Cross-contamination and the unexpected presence of gluten in processed foods or medications could increase the rate of involuntary gluten ingestion, in addition to the significant number of patients who do not adhere to chronic treatment [9]. Thus, in case of unresponsiveness to a GFD, whether intentional or unintentional, gluten ingestion should be considered as the main cause to check, and tests to confirm adherence to a GFD are necessary [1]. However, the evaluation of GFD correctness remains a difficult factor to achieve. Duodenal histology obtained during upper endoscopy provided endpoint information on the healing of the damage caused by the gluten intake, but it is invasive, and alteration of the small bowel is also reported in responsive patients [10]. Serological tests, such as for tTGA or deamidated gliadin peptide (DGP), are convenient methods for CD diagnosis but unreliable markers during the CD monitoring phase due to a high proportion of serology-negative patients in either cases of low adherence to a GFD or intestinal villus atrophy [11,12]. The celiac dietary adherence test (CDAT) is a non-invasive tool that allows a standardized assessment of the GFD adherence [13]. However, it is time-consuming, and the inputs are subjective and frequently subject to human mistakes, making it a low correlation with the duodenal atrophy that remains the reference standard to define CD activity [12,14]. 

Gluten fragments resistant to gastrointestinal digestion and capable of triggering CD are known as “gluten immunogenic peptides” (GIP) [15]. They can be absorbed in the gastrointestinal tract, can access blood circulation, and can be excreted in the urine. Several immunological methods using monoclonal antibodies (MoAbs) against GIP in urine (uGIP) and in feces (fGIP) have been developed to evaluate GFD adherence [16]. The uGIP lateral flow test is a point-of-care test that can usually detect a gluten ingestion of at least 50–500 mg that occurred during the previous 1–24 h [17,18]. Although some studies suggested that positivity with this test has shown a correlation with duodenal villus atrophy [11,15], its use in a real-life setting of an outpatient clinic remains persistently unclear as well as its efficacy in populations with different types of diets [19].

The aim of the study was to evaluate the role of the determination of uGIP test in a single sample of persistent CD outpatients and its correlation with other CD biomarkers previously used for the follow-up of celiac patients.

## 2. Methods

All CD patients referred to our tertiary referral Centre for Celiac Disease of the Fondazione IRCCS Ca’ Granda Ospedale Maggiore Policlinico of Milan, Italy from April 2019 to February 2020 were prospectively enrolled in the study. The CD diagnosis was posed according to international guidelines [1]. CD patients who reported intentional gluten ingestion at the follow-up visit and those following a GFD for less than six months were excluded from the study. All patients gave their written consent to be enrolled.

The presence of GIP in patients’ urine was assessed with the GlutenDetect^®^ test kit (Biomedal S.L., Seville, Spain), which detects the presence of GIP in urine, consequently to the ingestion of the protein complex in the preceding 12–24 h, depending on the individual and the type of gluten intake [17]. In the case of a uGIP positive test, the patients underwent a dietician control visit to verify the correctness of the GFD. 

Moreover, we evaluated socio-demographic data (sex, body mass index (BMI), age at enrolment, age at diagnosis), CDAT, patient health status by means of visual analogical scales (VAS), tTGA values, duodenal histology (graded following the Marsh–Oberhuber scale), and video capsule endoscopy (VCE) findings (PillCam™ SB3, Medtronic, Minneapolis, USA). In the case of VCE, the presence of atrophy and its extent was reported as validated in previous studies [14,20,21].

Briefly, CDAT is composed of seven questions about diet, each assessed on a 5-point scale. The overall score ranges from 7 to 35. Patients with higher scores denoted poorer adherence to a GFD. Values below 13 indicate good or excellent adherence to a GFD, while values equal to or above 17 indicate lower compliance [13].

VAS evaluated abdominal pain, satisfaction with stool consistency, bloating, postprandial fullness, early satiety, epigastric pain, and overall well-being.

### 2.1. Urinary Gluten Immunogenic Peptides Detection

All CD patients enrolled in the study underwent the GIP urinary test. They received a sterile urine container at the medical center on follow-up visits and were unaware of the test and the purpose of the urine collection. The urine samples were analyzed in the laboratory on the day of collection; the GlutenDetect kit (Biomedal, Seville, Spain) was used to test for the presence of uGIP. This methodology recognizes, in addition to the 33-mer peptide, similar peptides that react with the G12 and A1 monoclonal antibody, and it was used following the manufacturer’s instructions. Briefly, the test consists of a lateral flow-based strip, a separate tube containing conditioning solution, a disposable plastic syringe, and a quick guide with instructions for use. A urine sample is collected and transferred to the conditioning solution tube. The tube is tightly closed and gently shaken for 5 to 10 s. The lateral flow cassette is removed from its foil package, and 4 drops of the solution are added to the “S” zone on the cassette by unscrewing the transparent cap on the tube. After incubating for 15 min at room temperature, the cassette can be read [15]. 

### 2.2. Endoscopy and Histology

Upper gastrointestinal endoscopies and biopsies were performed with a high-definition gastroscope (Pentax, Toshima, Japan) and standard bioptic forceps (Boston Scientific, USA). Four oriented biopsy specimens were taken from the duodenum. Briefly, the samples were oriented on adhesive filter paper, fixed in a 10% formalin buffer, and paraffin-embedded. Sections were cut from each block and hematoxylin/eosin stained. Intraepithelial lymphocytes (IELs) were counted by means of anti-CD3 immunohistochemical staining (monoclonal mouse anti-human CD3 clone F7.2.38, Dako, Italy) [10]. Small bowel exploration was performed by means of video capsule endoscopy (VCE) (PillCam™ SB 3, Medtronic, Minneapolis, MN, USA). VCE was performed after intestinal cleaning with a 3-liter polyethylene-glycol (PEG)-based solution and overnight fasting. The Medtronic Imaging recording system was positioned according to the manufacturer’s instructions; data were downloaded on a dedicated computer workstation, analyzed by dedicated software, and read by an expert endoscopist [22,23].

CD patients undergoing upper endoscopy and VCE were selected following the international guidelines [1,24,25].

### 2.3. Statistical Analysis

All data were analyzed using percentages, mean, standard deviation (SD), median, and interquartile ranges. The distribution of the data was assessed with the Shapiro–Wilk test. Quantitative variables with a normal distribution (age at CD diagnosis, age at enrolment, BMI, CDAT score) were compared between the two groups using the students’ two-tailed *t*-test for independent variables assuming two different variances; for the qualitative variables, association analyses were performed between the independent variables (positive/negative test) and the dependent variables (sex, age, anti-tTG, atrophy at Marsh–Oberhuber biopsy, atrophy at VCE) and compared using the chi-squared test or Fisher’s exact tests if more than 20% of the cells had a predicted count of less than five. When the chi-squared test was used, the correlation coefficient Phi or Cramer’s V was also evaluated to assess how independent the variables were (0 = independence, 1 = dependence). The significance level α was set at 0.05. 

The statistical analysis was performed using Microsoft Excel^®^ version 2019 (Microsoft Corporation, Redmond, WA, USA) and the Statistical Package for the Social Sciences (SPSS) version 20.0 (IBM SPSS Statistics for Windows, Version 20.0; IBM Corp., New York, NY, USA).

The study was approved by the local Ethics Committee Name (Milano area 2), approval code 19_2019bis.

## 3. Results

### 3.1. Patients, uGIP Results, and Symptom Assessment

A total of 285 CD patients were enrolled in the study. Among them, 113 (39.6%) performed an upper endoscopy with duodenal biopsies, and 61 (21.4%) underwent VCE with different indications (refractory coeliac disease, alarm symptoms and suspicion of complications, persistence of serological positivity). Among the 285 patients, five did not complete the study and were excluded from the analysis. Therefore, 280 patients, including 232 (82.8%) females and 48 (17.2%) males with a median age of 31.7 ± 16.9, completed the prospective study.

Table 1 shows the clinical and demographic data of the patients in association with the uGIP results. No association was found between uGIP findings and sex, age at diagnosis and enrollment, BMI, and CDAT. However, in patients with positive uGIP, significantly higher severity of abdominal bloating was reported (*p* < 0.05) (Table 2).

### 3.2. Correlation between Anti-Tissue Transglutaminase Antibodies, Small Bowel Atrophy, Video Capsule Endoscopy, and uGIP Findings

Among the 280 celiac patients enrolled, 69 (24.6%) had a tTGA titre above the threshold of normality. In the group with positive uGIP patients, 10 (31.2% of the 32 uGIP+) were positive for tTGA vs. 22 (68.8% of the uGIP+) negative for tTGA, without a statistically significant difference. Furthermore, the tTGA+ titre was not related to the positivity of uGIP (14.4% vs. 10.9% in patients with tTGA+ and tTGA−). Again, no clinical or demographic differences were found between tTGA-positive and negative patients.

A total of 113 patients underwent upper endoscopy with duodenal biopsies. Among them, 71 (62.8%) patients did not show atrophy (i.e., grade 0, 1, or 2 according to the Marsh–Oberhuber scale), and 5 (7.0%) were uGIP positive. Among the 42 (37.2%) patients with duodenal atrophy (Marsh–Oberhuber 3a, 3b, 3c), ten (23.1%) presented a positive uGIP test (*p* < 0.05, Figure 1). The tTGA titre did not show a significant correlation either with duodenal atrophy (Table 3) or with uGIP (see above).

Among the patients enrolled, 61 patients underwent VCE: 32 (52.4%) patients did not present any endoscopic sign of mucosal atrophy, and four (12.5%) were positive for uGIP; 29 (47.5%) VCE showed endoscopic signs of mucosal atrophy and five (17.2%) were positive for the uGIP test without a statistical difference, compared to unremarkable VCE. Among VCE, 18 (29.5%) presented an extension of mucosal atrophy for more than 30% of the recording time, and four (22.2%) had a positive uGIP compared to five (11.6%) in the case of no or proximal small bowel atrophy (no statistical difference). Distribution of atrophy at VCE in uGIP+ and uGIP- patients is reported in Appendix A.

In Figure 2 tTGA, the mucosal atrophy of both is summarised. Notably, in the uGIP+ group, 66.7% of patients presented duodenal atrophy, but 21.4% did not present any indirect biomarker (mucosal atrophy or tTGA) suggesting the ingestion of gluten. Conversely, in the uGIP− group, 32.7% presented duodenal atrophy and roughly half the patients did not present markers suggesting gluten ingestion. Again, the tTGA result did not differ between uGIP+ and uGIP− patients and did not correlate with atrophy in both groups. 

## 4. Discussion

Our study revealed that 11% of CD patients attending an Italian tertiary referral center, unaware of the test, had a positive uGIP although they declared complete adherence to the GFD. A positive uGFD did not correlate with symptoms, CDAT, or tTGA values, but more frequently duodenal atrophy was observed in patients with a detectable amount of GIP in urine. 

Actual guidelines suggest that compliance with the GFD should be strict in most cases and voluntary gluten intake (transgressions) is not allowed. The GFD compliance rate could vary between 25% and 89% depending on the age, sex, time of observation, and the minimal amount of gluten to be assumed as gluten exposure [26,27,28]. Several factors could be associated with good GFD adherence: good knowledge of the disease, its treatment and permitted foods, a high level of education, good social and economic status, female sex, young age, high self-esteem, and a willingness to schedule regular specialist visits [29,30,31,32]. The necessary attention towards contamination can be undermined by the practical, social, and psychological restrictions imposed by the GFD.

All these reasons make GFD monitoring a central issue in the management of CD patients, especially when facing a poor clinical, serological, or histological response to the diet. The individuation of subjects with involuntary ingestion of gluten is important to implement the health status of celiac patients. 

The recent introduction of uGIP makes the direct control of a GFD possible. The uGIP is a simple and reliable point-of-care test that is able to detect the ingestion of gluten during the last 12–24 h. In our group, CD patients were unaware of the test, and 11% showed the presence of GIP in urine demonstrating *de facto* an unconscious gluten ingestion during the last 24 h before the visit. This finding is not very far from those previously reported by Ruiz-Carnicer et al. [11], who reported 21% of positive urine GIP with the sample on the day of the visit, although the positivity rate increased over the weekend (30–36%). In this study, the sensitivity of the method to predict mucosal damage was 39% for a single test on the day of the visit and a specificity of 84%. In our study, positive uGIP predicts villus atrophy with a high specificity of 93%. Notably, our patients were unaware of the test, and thus they probably did not change their diet habits the day before the clinical control. Gluten content in food is regulated by the Codex Alimentarius [33], which since 2008 has stated that products can be defined as gluten-free if they are naturally gluten-free (e.g., rice, maize, potatoes) or if they contain an amount less than or equal to 20 ppm (parts per million). Several contaminations could occur in everyday life, and nutritionist evaluation maintains its central role in maintaining patient adherence to a GFD and correct nutritional intake. However, the relationship between the amount of gluten ingested and the development of symptoms is not clearly defined, and the exact amount of gluten that people with CD can tolerate daily without experiencing deleterious effects has not been fully established, nor is it yet clear how strict a GFD should be [34]. Celiac patients usually avoid consuming any products that potentially contain gluten, thus eliminating many common foods from their diet. This can have both psychological consequences as it leads to isolation and health consequences as specific nutritional deficiencies may develop. The appropriate approach to the GFD is therefore a balance between the strict exclusion of gluten-containing products and nutritional adequacy [35,36,37]. A clinical assessment of CD patients is usually required and supported by questionnaires assessing adherence to the self-reported GFD, such as the CDAT [13]. Our studies show no correlation between CDAT and positivity to uGIP, suggesting that questionnaires could miss “contaminated” patients. Similarly, serology markers (tTGA) could not reflect either gluten ingestion revealed by uGIP or intestinal mucosal damage in most patients with persistent villus atrophy. All autoantibodies related to CD are expected to decrease from baseline after months of strict adherence to the GFD [28]. These findings are in line with those previously reported by authors, who claim a limited role for serological tests as a surrogate to evaluate adherence to a GFD, as most CD patients with positive fGIP or uGIP were negative in serology [12,38,39,40]. 

There is much debate about follow-up biopsies because in adults neither symptoms nor serology is a reliable predictor of small bowel damage [38,39]. At present, there are no studies indicating an absolute need for routine follow-up biopsies for all patients. However, those celiac patients with the most frequent or high gluten ingestions reported by fGIP or uGIP have turned out to be mostly asymptomatic and seronegative [11,12]. In addition, symptomatic patients are given a biopsy to exclude refractory CD or malignancy. Therefore, a follow-up biopsy could be necessary for patients with serological negative CD because this is the only possible way so far to confirm response to a GFD and is considered the reference standard to define CD activity. In particular, uGIP positivity correlates with duodenal atrophy at histology, demonstrating a certain CD activity. However, this activity is confined to the proximal part of the small bowel, as demonstrated by VCE analysis, here reported for the first time in the literature. 

Notably, about 25% of patients with positive uGIP did not present any positivity of tTGA or signs of histological damage. Although one single uGIP test could be of limited statistical significance in the habit or sensitivity of CD patients, a recent study evaluated the possibility that a certain percentage of patients could tolerate occasional ingestion of gluten [41]. From this point of view, the surveillance of uGIP could become relevant for future studies on this issue. 

Different limitations of the study have to be explored; all patients were not equally evaluated, and a single urine sample was taken. As a consequence, gluten intake could be under-evaluated, since multiple samples, covering the weekend as well as weekdays, can increase sensitivity and reduce the loss of positive cases [42].

## 5. Conclusions

uGIP may play an important role as a point-of-care test to be undertaken by patients themselves or a physician during monitoring (as in the present study). The test has been demonstrated as a useful tool to evaluate adherence to the GFD and/or accidental gluten contamination.

## Figures and Tables

**Figure 1 nutrients-15-01259-f001:**
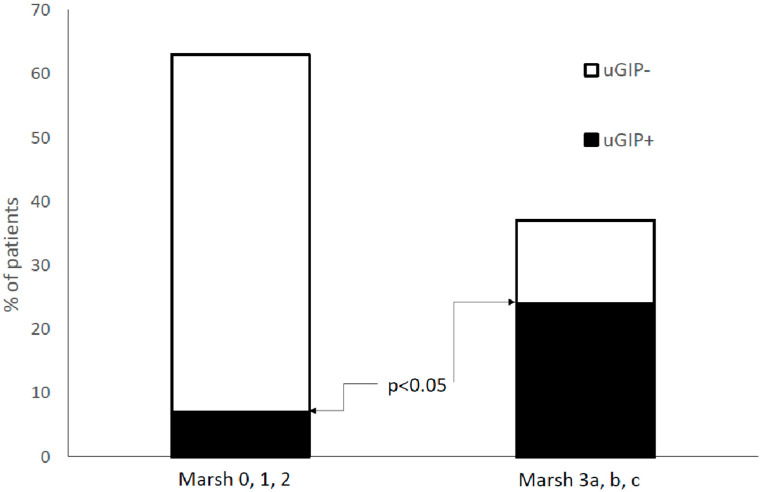
Rate of celiac patients with or without duodenal atrophy and with or without positive urinary gluten immunogenic peptides (uGIP).

**Figure 2 nutrients-15-01259-f002:**
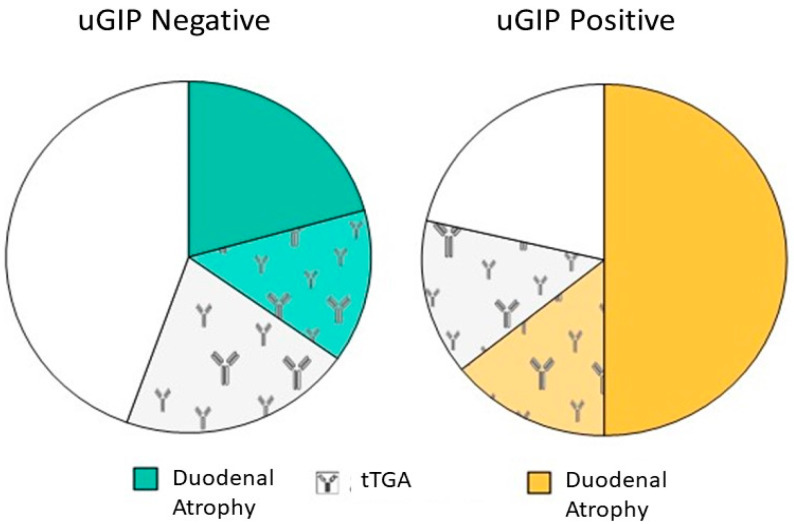
Duodenal atrophy and tTGA positivity in celiac patients with or without a uGIP positive test.

**Table 1 nutrients-15-01259-t001:** Clinical and demographic characteristics of celiac patients with positive or negative urinary gluten immunogenic peptides (uGIP) results.

Total		*uGIP*		*p*
		Negative (−)	Positive (+)	
Sample, *n* (%)	280	248 (88.6)	32 (11.4)	-
Sex F, *n* (%)	232 (82.8)	204 (82.2)	28 (87.5)	0.4
Age at diagnosis of CD	31.7 ± 16.9	31.4 ± 16.9	33.6 ± 17.5	0.5
Age at enrolment	42.9 ± 15.2	42.62 ± 15.0	45.66± 16.3	0.3
GFD (years)	11.1 ± 10.2	11.1 ± 10.2	11.4 ± 9.9	0.8
BMI, kg/m²	22.2 ± 3.9	22.2 ± 4.0	22.1 ± 3.3	0.8
CDAT score	13.8 ± 2.6	13.8 ± 2.4	14.2 ± 4.2	0.6

Note: CD—celiac disease; BMI—body mass index; CDAT—celiac dietary adherence test; GFD—gluten-free diet.

**Table 2 nutrients-15-01259-t002:** Symptoms reported by visual analogue scales (VAS) in celiac patients with positive and negative uGIP.

		*uGIP*		*p*
Symptom	Total	Negative (−)	Positive (+)	
	280	248	32	-
Abdominal pain	1.0 (4.0)	1.0 (4.0)	1.0 (4.8)	0.97
Satisfaction with stool consistency	7.0 (6.0)	6.5 (5.8)	7.5 (8.0)	0.81
Abdominal bloating	3.0 (7.0)	3.0 (7.0)	5.5 (6.5)	0.03
Postprandial fullness	2.0 (7.0)	2.0 (6.8)	4.5 (8.8)	0.34
Premature satiety	1.0 (5.0)	1.0 (5.0)	1.0 (4.0)	0.84
Epigastric burning	0.5 (4.0)	1.0 (10.0)	0.0 (4.8)	0.70
Overall well-being	8.0 (3.0)	8.0 (3.0)	8.0 (4.8)	0.40

Note: uGIP: gluten immunogenic peptides in urine. data expressed as median and interquartile range; Mann–Whitney U test.

**Table 3 nutrients-15-01259-t003:** Anti-tissue transglutaminase immunoglobulin A (tTGA) and atrophy contingency table.

		tTGA		Total
		Negative (−)	Negative (+)	
Atrophy (Marsh–Oberhuber)	No, *n* (%)	45 (61.4)	23 (57.5)	68 (60.1)
	Yes, *n* (%)	28 (38.6)	17 (42.5)	45 (39.9)
Total, *n* (%)		73 (100)	40 (100)	113 (100)

Note: Data expressed as absolute frequency; Pearson’s chi-square test: *p* = 0.65.

## Data Availability

Data sharing is not applicable to this article.

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
