# Peer review of "Evaluation of a Single Determination of Gluten Immunogenic Peptides in Urine from Unaware Celiac Patients to Monitor Gluten-Free Diet Adherence"

_nutrients, 2023, doi:10.3390/nu15051259_

Round 1

Reviewer 1 Report

Brief summary

I thank the authors for their contribution with this paper. It is an article that values the importance of other methods in the follow-up of Celiac Disease, because it is known that adherence to the diet is not easy and that dietary transgressions are frequent.

Broad comments

Title. It is monitored if there is gluten, not if it is conscious or unconscious, it may be that the patient deceives the doctor, so I suggest eliminating “unconscious” and changing  by due to adherence to a gluten-free diet or presence of gluten intake

I do not like to include this question in the title, the potential tolerance to gluten is yet to be determined and it is not in accordance with the objective of the study, I suggest eliminating it.

Abstract. the biopsy would not be an indirect method, but it would be a very invasive method

Introduction. I suggest including the reference indicated below, which evaluates the different methods for monitoring patients, and which assesses the need for new methods, such as stool and urine GIPs.

·         Gerasimidis K, Zafeiropoulou K, Mackinder M, Ijaz UZ, Duncan H, Buchanan E, et al. Comparison of clinical methods with the faecal gluten immunogenic peptide to assess gluten intake in coeliac disease. J Pediatr Gastroenterol Nutr. 2018;67:356-60.

Material and methods. Indicate what were the criteria for some patients to have certain techniques and others to undergo other techniques.

Results. Table 1 should better include the baseline characteristics of the disease, such as the time on the gluten-free diet and whether there were differences based on the time of adherence to the diet. It must also be included when the biopsy was performed, since it is not known how long the patients who were positive had been with GFD. It would be important to indicate that as well. Was it because any of the patients with a positive biopsy had been on a gluten-free diet for less than 12-18 months, which is the time cited by some cohorts to normalize histology? It is necessary to clarify it.

Regarding sex, to what do the authors attribute the striking differences between the sexes and much higher than the differences reported in the literature?  (Prevalence in women is higher than men but not so much)

Discussion. It is important to indicate limitations of the study. 1) a single urine sample was taken, which may mean that gluten intake has not been adequately evaluated, since it is appropriate to take a sample that covers the weekend, as well as weekdays, in order to gain sensitivity with the test, therefore, there could have been a loss of positive cases for this reason. 2) All patients were not equally evaluated.

Specific comments

Line 41. Text is missing to complete the EMA and HLA sentence.

Line 152. Please indicate normal laboratory values

Line 189: the authors must cite the statement they make about the consequences of poor adherence to the gluten-free diet.

Line 209: The authors assume that consumption is unconscious, but it may not be.

Line 227-228: please, include the reference.

Line 256. I do not agree in considering gluten consumption "safe" for celiac patients, simply because there is no correlation with symptoms or intestinal damage, since it has not been evaluated from other points of view that are being included in research, such as transcriptomics, metabolomics, etc. Therefore, I would delete the last sentence, urine GIPS does not help to manage gluten-tolerant patients.

Author Response

REVIEWER 1

Comments and Suggestions for Authors

Brief summary

I thank the authors for their contribution with this paper. It is an article that values the importance of other methods in the follow-up of Celiac Disease, because it is known that adherence to the diet is not easy and that dietary transgressions are frequent.

 Broad comments

- Title. It is monitored if there is gluten, not if it is conscious or unconscious, it may be that the patient deceives the doctor, so I suggest eliminating “unconscious” and changing  by due to adherence to a gluten-free diet or presence of gluten intake. I do not like to include this question in the title, the potential tolerance to gluten is yet to be determined and it is not in accordance with the objective of the study, I suggest eliminating it.

Thank you for your suggestion, the title has been modified.

- Abstract. the biopsy would not be an indirect method, but it would be a very invasive method

      Thank you for your suggestion, the text has been modified at line 21.

- Introduction. I suggest including the reference indicated below, which evaluates the different methods for monitoring patients, and which assesses the need for new methods, such as stool and urine GIPs.

  • Gerasimidis K, Zafeiropoulou K, Mackinder M, Ijaz UZ, Duncan H, Buchanan E, et al. Comparison of clinical methods with the faecal gluten immunogenic peptide to assess gluten intake in coeliac disease. J Pediatr Gastroenterol Nutr. 2018;67:356-60.

      Thank you for your suggestion, the reference has been added at line 81.

- Material and methods. Indicate what were the criteria for some patients to have certain techniques and others to undergo other techniques.

Patients underwent endoscoy and enteroscopy following the indications described in the intrenational guidelines

- Results. Table 1 should better include the baseline characteristics of the disease, such as the time on the gluten-free diet and whether there were differences based on the time of adherence to the diet. It must also be included when the biopsy was performed, since it is not known how long the patients who were positive had been with GFD. It would be important to indicate that as well. Was it because any of the patients with a positive biopsy had been on a gluten-free diet for less than 12-18 months, which is the time cited by some cohorts to normalize histology? It is necessary to clarify it.

Accordingly, in table 1 the GFD timelapse has been added. In materials and methods, the time between uGIP and endoscopy has been specified (maximum 2 months)

- Regarding sex, to what do the authors attribute the striking differences between the sexes and much higher than the differences reported in the literature?  (Prevalence in women is higher than men but not so much)

women are usually more inclined to be monitored and undergo control visits

- Discussion. It is important to indicate limitations of the study. 1) a single urine sample was taken, which may mean that gluten intake has not been adequately evaluated, since it is appropriate to take a sample that covers the weekend, as well as weekdays, in order to gain sensitivity with the test, therefore, there could have been a loss of positive cases for this reason. 2) All patients were not equally evaluated.

Thank you for your suggestion, we have added the sentence at line 262-266.

Specific comments

- Line 41. Text is missing to complete the EMA and HLA sentence.

The text has been implemented.

- ??? Line 152. Please indicate normal laboratory values

Added

- Line 189: the authors must cite the statement they make about the consequences of poor adherence to the gluten-free diet.

The sentence has been added

- Line 209: The authors assume that consumption is unconscious, but it may not be.

Thank you for your suggestion, the sentence has been modified at line 216.

- Line 227-228: please, include the reference.

As suggested, the reference has been included.

- Line 256. I do not agree in considering gluten consumption "safe" for celiac patients, simply because there is no correlation with symptoms or intestinal damage, since it has not been evaluated from other points of view that are being included in research, such as transcriptomics, metabolomics, etc. Therefore, I would delete the last sentence, urine GIPS does not help to manage gluten-tolerant patients.

As suggested, the last sentence has been deleted.

Reviewer 2 Report

This work assessed the actual adherence of the gluten-free diet in celiac patients by measuring the level of the urinary gluten immunogenic peptides (uGIP) and determined the correlation of this detection with other biomarkers of celiac disease by combining the diagnosis of celiac disease and dietary adherence-related indicators. This study was based on a series of assays on human samples. However, this research was not sufficient to demonstrate a strong link between uGIP detection and patient adherence to the gluten-free diet, and the association of uGIP with celiac disease-related biomarkers was not fully sorted out. Specific comments and questions are as follows:

1. The main article type should be Article rather than Review.

2. The "Methods" section of the Abstract was not sufficiently described. Moreover, the relevant data in the Abstract section were not found in the manuscript (such as 14.4%, 10.9%, 66.7%, 32.7%, 24 GIP- and 5 GIP+), the details should be presented in the Results section.

3. 11% or 11.4% should be expressed consistently in the manuscript.

4. Some parts of the introduction lacked references, such as Line 46-51 and 71-75.

5. The Methods were missing the detailed description of the patient tests, such as how soon to inform the patient before the urine collection, and the method of urine collection. Furthermore, the determination of various indicators in Line 94-99 missed the introduction of reagents or instruments.

6. In the Results section, Line 151-155 and Line 167-173  lacked the corresponding data or graphical support, please add them in the supplementary materials.

7. The presentation of the results in Line 160-161 was not complete, for example, the comparison of different tTGA titers at the same Atrophy.

8. Why was the Total in Table 3 106 instead of 113?

9. What were the specific data shown in Figure 2, such as the percentage of each item?

10. Line 183-205 should be streamlined and placed in the introduction rather than in the discussion section.

11. Line 215-221 should be streamlined and placed in the methods section.

12. Line 221-228 should not be part of the discussion section. The Discussion section should focus on the analysis of the data results, the comparison with the latest research, and its own limitations. It is suggested that the discussion section should be rewritten.

13. The Institutional Review Board Statement should be supplemented.

14. The content of "(and potential tolerance to gluten?)" in the title of the paper should be removed.

Author Response

REVIEWER 2

Comments and Suggestions for Authors

This work assessed the actual adherence of the gluten-free diet in celiac patients by measuring the level of the urinary gluten immunogenic peptides (uGIP) and determined the correlation of this detection with other biomarkers of celiac disease by combining the diagnosis of celiac disease and dietary adherence-related indicators. This study was based on a series of assays on human samples. However, this research was not sufficient to demonstrate a strong link between uGIP detection and patient adherence to the gluten-free diet, and the association of uGIP with celiac disease-related biomarkers was not fully sorted out. Specific comments and questions are as follows:

  1. The main article type should be Article rather than Review.

It has been modified.

  1. The "Methods" section of the Abstract was not sufficiently described. Moreover, the relevant data in the Abstract section were not found in the manuscript (such as 14.4%, 10.9%, 66.7%, 32.7%, 24 GIP- and 5 GIP+), the details should be presented in the Results section.
  2. 11% or 11.4% should be expressed consistently in the manuscript.

The findings have been indicated in results section

  1. Some parts of the introduction lacked references, such as Line 46-51 and 71-75.

As suggested, references has been added.

  1. The Methods were missing the detailed description of the patient tests, such as how soon to inform the patient before the urine collection, and the method of urine collection. Furthermore, the determination of various indicators in Line 94-99 missed the introduction of reagents or instruments.

Patients were completely unaware about the test and its significance

  1. In the Results section, Line 151-155 and Line 167-173  lacked the corresponding data or graphical support, please add them in the supplementary materials.
  2. The presentation of the results in Line 160-161 was not complete, for example, the comparison of different tTGA titers at the same Atrophy.
  3. Why was the Total in Table 3 106 instead of 113?
  4. What were the specific data shown in Figure 2, such as the percentage of each item?

Results section has been modified accordingly to the suggestions

  1. Line 183-205 should be streamlined and placed in the introduction rather than in the discussion section.

Thank you for your suggestion. As suggested, the paragraph has been inserted in the introduction.

  1. Line 215-221 should be streamlined and placed in the methods section.

Thank you for your suggestion. As suggested, the paragraph has been inserted in the introduction.

  1. Line 221-228 should not be part of the discussion section. The Discussion section should focus on the analysis of the data results, the comparison with the latest research, and its own limitations. It is suggested that the discussion section should be rewritten.

Accordingly, this part of the manuscript has been corrected

  1. The Institutional Review Board Statement should be supplemented.

Thank you for your suggestion, the information has been added in the methods at line 107-111.

  1. The content of "(and potential tolerance to gluten?)" in the title of the paper should be removed.

Thank you for your suggestion, the sentence has been removed.

Reviewer 3 Report

Dear authors,

This is an interesting manuscript introducing the test that could make direct control of gluten free diet possible. However, the manuscript should be improved.

Firstly, the language needs improvement. 

Lines 39 - 42: Please rewrite the sentence where you describe criteria for the diagnosis of celiac disease. Here you mix everything from villus atrophy to autoantibodies and genetics.. Can you rewrite the sentence for better clarity. 

Line 53 - 56. The sentence needs corrections. It is not clear.

What is GFD correctness?

In methods you wrote who was excluded from the study but not who was included in the study. Which is imprtant.

Lines 106-108: Please rewrite. This sentence is not ok.

Is the legend of figure 2 ok? I do not understand it. Please describe the figure in few sentences in the result section.

The discussion has a lot of theory which could be moved into introduction but lacks more discussion on results.

Conclusion should be given based on the results if the research.

References are appropriate.

Author Response

REVIEWER 3

Comments and Suggestions for Authors

Dear authors,

This is an interesting manuscript introducing the test that could make direct control of gluten free diet possible. However, the manuscript should be improved.

- Firstly, the language needs improvement. 

- Lines 39 - 42: Please rewrite the sentence where you describe criteria for the diagnosis of celiac disease. Here you mix everything from villus atrophy to autoantibodies and genetics.. Can you rewrite the sentence for better clarity. 

Thank you for your suggestion, the sentence has been modified.

- Line 53 - 56. The sentence needs corrections. It is not clear.

Thank you for your suggestion, the sentence has been modified.

- What is GFD correctness?

It is the correct adherence. The text has been changed.

- In methods you wrote who was excluded from the study but not who was included in the study. Which is important.

Methods section has been implemented

- Lines 106-108: Please rewrite. This sentence is not ok.

Thank you for your suggestion, the sentence has been modified.

- Is the legend of figure 2 ok? I do not understand it. Please describe the figure in few sentences in the result section.

Accordingly, the results section has been implemented

- The discussion has a lot of theory which could be moved into introduction but lacks more discussion on results.

- Conclusion should be given based on the results if the research.

Accordingly, the discussion-conclusion section has been implemented

Round 2

Reviewer 2 Report

The following issues remain in the current revised version, which will seriously affect the quality of the article.

1. Question 6 has not been resolved.

2. The decimal points of the values in the article should be uniform, which could be rounded or retained to one valid digit after the decimal point.

3. Questions 10, 11, and 12 are not resolved.

Author Response

We thank the referee for the interest in our study. In the first and second revision we tried to answer to all the suggestions of the three reviewers and accomplish with the editorial concerns and editing rules. Thus, we did our best to answer to all the questions (sometime conflicting) and tried to choose the best solution among different ones

Question 6 has not been resolved

Line 151-155: data are presented in table 1

line 167-173:  A supplementary figure showing the distribution of atrophy along the small bowel of GIP+ and GIP- patients has been added

The decimal points of the values in the article should be uniform, which could be rounded or retained to one valid digit after the decimal point

Values have been harmonized

Questions 10, 11, and 12 are not resolved

Line 183-196 have been moved to introduction, we preferred to leave the other lines in the discussion

lines 215-221 explain the regulatory about gluten content and underline that patients were unaware about the test, furthermore methods section has completely rewritten

lines 221-228 discuss the the possible reasons why a number of patients with positive test do present an active CD. We prefer to leave it in the discussion to not unbalance the MS

Reviewer 3 Report

I checked the corrections made and agree with the publication of manuscript in the present form.

Author Response

Thank you